# Hydrolysis of Edible Oils by Fungal Lipases: An Effective Tool to Produce Bioactive Extracts with Antioxidant and Antimicrobial Potential

**DOI:** 10.3390/foods11121711

**Published:** 2022-06-10

**Authors:** Alexandra Kotogán, Zsófia Terézia Furka, Tamás Kovács, Bettina Volford, Dóra Anna Papp, Mónika Varga, Thu Huynh, András Szekeres, Tamás Papp, Csaba Vágvölgyi, Keshab Chandra Mondal, Erika Beáta Kerekes, Miklós Takó

**Affiliations:** 1Department of Microbiology, Faculty of Science and Informatics, University of Szeged, Közép fasor 52, H-6726 Szeged, Hungary; primula15@gmail.com (A.K.); furka.zsofi@gmail.com (Z.T.F.); kovacstomi16@gmail.com (T.K.); bettina.volford86@gmail.com (B.V.); papp.dori.anna@gmail.com (D.A.P.); varga.j.monika@gmail.com (M.V.); huynh_thu@hcmut.edu.vn (T.H.); szandras@bio.u-szeged.hu (A.S.); pappt@bio.u-szeged.hu (T.P.); csaba@bio.u-szeged.hu (C.V.); kerekeserika88@gmail.com (E.B.K.); 2Department of Microbiology, Vidyasagar University, Midnapore 721102, India; mondalkc@gmail.com

**Keywords:** enzyme-assisted hydrolysis, microbial lipases, vegetable oils, menhaden fish oil, bioactive fatty acids, antioxidant and antimicrobial activities

## Abstract

Hydrolysis of olive, rapeseed, linseed, almond, peanut, grape seed and menhaden oils was performed with commercial lipases of *Aspergillus niger*, *Rhizopus oryzae*, *Rhizopus niveus*, *Rhizomucor miehei* and *Candida rugosa*. In chromogenic plate tests, olive, rapeseed, peanut and linseed oils degraded well even after 2 h of incubation, and the *R. miehei*, *A. niger* and *R. oryzae* lipases exhibited the highest overall action against the oils. Gas chromatography analysis of vegetable oils hydrolyzed by *R. miehei* lipase revealed about 1.1 to 38.4-fold increases in the concentrations of palmitic, stearic, oleic, linoleic and α-linolenic acids after the treatment, depending on the fatty acids and the oil. The major polyunsaturated fatty acids produced by *R. miehei* lipase treatment from menhaden oil were linoleic, α-linolenic, hexadecanedioic, eicosapentaenoic, docosapentaenoic and docosahexaenoic acids, with yields from 12.02 to 52.85 µg/mL reaction mixture. Folin–Ciocalteu and ferric reducing power assays demonstrated improved antioxidant capacity for most tested oils after the lipase treatment in relation to the concentrations of some fatty acids. Some lipase-treated and untreated samples of oils, at 1.25 mg/mL lipid concentration, inhibited the growth of food-contaminating bacteria. The lipid mixtures obtained can be reliable sources of extractable fatty acids with health benefits.

## 1. Introduction

Fatty acids are important bioactive molecules and can play a substantial role in the prevention and management of several diseases. For instance, polyunsaturated fatty acids (PUFAs) and conjugated isomers of unsaturated fatty acids can lower the risks for chronic diseases such as diabetes and atherosclerosis [1,2]. Saturated short-chain fatty acids, i.e., acetate, butyrate and propionate, can modulate inflammation by inhibition or induction of the production of various regulatory cytokines (e.g., IL-2 and interferon-γ) [3].

The edible oils, including vegetable and fish oils, contain a wide variety of bioactive fatty acids that can be used as functional additives in foods after extraction [4]. Different edible oils have different fatty acid compositions; most of them are present as triacylglycerols. For instance, the main fatty acid composition of olive oil is palmitic acid (PA; C16:0; 7.5–20%), palmitoleic acid (PLA; C16:1; 0.3–3.5%), stearic acid (SA; C18:0; 0.5–5%), oleic acid (OA; C18:1; 55–83%), linoleic acid (LA; C18:2; 3.5–21%) and α-linolenic acid (ALA; C18:3; 0–1.5%) [5]. Grape seed oil is characterized by monounsaturated (e.g., OA, 20–40%) and saturated (around 10%) fatty acids; and its PUFA content is about 85–90%, which mostly comprises LA (65–75%) [6,7,8]. OA (37.2–61.7%), LA (23.4–43.1%) and PA (8.8–13%) are the main fatty acid constituents in peanut oils, depending on the variety of peanut studied [9]. Fatty acids constituents such as OA (64–82%), LA (8–28%) and PA (6–8%) can be found in almond oil in high quantities [10]. The overall composition of rapeseed oil was characterized by 2.1–5.1% PA and 0.7–2.6% SA content, and by a variety of unsaturated fatty acids, such as OA (8.4–64.6%), LA (11.2–22.3%) and ALA (4.8–22.8%), depending on the *Brassica* species used for oil extraction [11]. Linseed oil is rich in ALA (39.9–60.4%), LA (12.2–17.4%), OA (13.4–19.3%), SA (2.2–4.5%) and PA (4.9–8.0) [12]. Consumption of these vegetable oils can reduce the risks of developing cardiovascular diseases, diabetes, cancer and/or arthritis; and numbers of them are used as food supplements due to their health benefits [6,7,10,12,13,14,15]. Among the fish oils, the bioactive properties of menhaden oil are among of the most intensively studied. The commercial oil is mostly extracted from Atlantic menhaden (*Brevoortia tyrannus*) or Gulf menhaden (*Brevoortia patronus*), and is rich in omega-3 PUFAs, i.e., eicosapentaenoic acid (EPA; C20:5; 11.1–16.3%) and docosahexaenoic acid (DHA; C22:6; 4.6–13.8%), which can reduce the risk of cardiovascular diseases [16,17]. Other important fatty acid components of menhaden oil triglycerides are PA (15.3–25.6%), PLA (9.3–15.8%) and OA (8.3–13.8%) [16].

The antimicrobial capacities of various plant and animal oils [7,18] and individual fatty acids [1,19,20,21] are also known. The effect of unsaturated fatty acids against foodborne pathogens has been studied heavily [22,23]. In addition, the effects of LA, AA, EPA and DHA have been demonstrated against *Plasmodium* species [24], and antiviral activity has also been documented for some compounds [25]. Taken together, these natural fatty acid molecules can be excellent supports for conventional preservatives in food and cosmetic products. The individual fatty acids can also possess antioxidant properties in certain reactions [26,27,28]. For instance, a recent study reported a significant association between the contents of some fatty acids and the total antioxidant capacities of grape pomace samples [29]. However, the major antioxidants and antimicrobials in most edible oils are the polyphenolic and phenol ester compounds [30,31].

The majority of bioactive fatty acids in natural oils are usually bounded in triglycerides, which reduces their bioavailability [32]. These compounds can be obtained from the oil through sub- and supercritical water extraction [33], or acidic, alkaline or enzyme-assisted hydrolysis reactions [34,35,36]. The lipase-assisted hydrolysis has reported to be as an ecofriendly approach for the extraction. Lipases (EC 3.1.1.3) catalyze the hydrolysis of triglycerides in natural oils, generating free fatty acid molecules from the lipid substrate. The hydrolytic activity of these biocatalysts is widely utilized in the food processing, and in the pharmaceutical, detergent and cosmetic industries [37]. Microorganisms are excellent sources of lipases, and enzymes from certain fungi, such as *Aspergillus*, *Candida*, *Rhizomucor* and *Rhizopus* species, are commercially available and frequently used in food applications [38]. The potential of fungal lipases to enrich fatty acids from different oils, e.g., soybean, olive, castor, palm, coconut safflower, linseed, rapeseed and salmon, has been documented [35,39,40,41,42,43,44,45]. However, less attention has been paid to the bioactive properties, i.e., antioxidant and antimicrobial capacities, of the produced fatty-acid-enriched samples.

In this study, an attempt was made to produce fatty-acid-enriched extracts with bioactive properties from olive, rapeseed, linseed, almond, peanut, grape seed and menhaden fish oils by lipase-assisted extraction. To analyze the hydrolytic efficiency of the fungal lipases involved, firstly, lipolysis of oils was screened via a chromogenic plate method; then, released fatty acids were determined in liquid reaction mixtures set up by selected enzymes. After the enzyme treatment, bioactive properties of the enzyme-free and treated natural oils, i.e., antioxidant capacities and effects against food-contaminating bacteria, were examined and compared. In addition, pure forms of some selected fatty acids were also subjected to antimicrobial analysis for comparison.

## 2. Materials and Methods

### 2.1. Sources of Lipases, Oils and Fatty Acids

Hydrolysis reactions were performed by *Aspergillus niger* (product number 62301), *Candida rugosa* (product number 62316), *Rhizopus oryzae* (product number 62305), *Rhizopus niveus* (product number 62310) and *Rhizomucor miehei* (formerly known as *Mucor miehei*, product number 62298) free lipases obtained from Sigma–Aldrich (Munich, Germany). Seven oils, i.e., olive (refined; product number O1514), linseed (product number 430021), peanut (product number P2144), grape seed (product number W233218), rapeseed (product number 83450), almond (product number 63445) and menhaden fish (refined; product number F8020) oils, were involved to the assay. All oils were purchased from Sigma–Aldrich (Munich, Germany). PA (5.30–25.23%) was a major saturated fatty acid component in the oils used as determined by gas chromatography (GC) [46]. Concerning unsaturated fatty acids, OA (5.12–80.05%) and LA (1.91–23.79%) were among the major components of the oils. In addition, ALA (58.33%) in linseed oil and EPA (12.11%) and DHA (5.47%) in menhaden fish oil were also predominant fatty acid components. Individual fatty acids, i.e., EPA and ALA, were obtained from VWR (Debrecen, Hungary).

### 2.2. Preparation of Enzyme Solutions

Solutions of 0.1, 1 and 10 mg/mL of each enzyme were prepared in phosphate buffer (50 mM, pH 7.4). To prepare a control sample with inactivated enzyme, a small volume from the 10 mg/mL stock solution was placed into boiling water for 5 min.

### 2.3. Lipase Activity Assay

Lipase activity of the enzyme solutions was determined following a standard spectrophotometric method [47] based on the reaction against the substrate *p*-nitrophenyl palmitate (*p*NPPa; Sigma–Aldrich, Munich, Germany). To dilute the *p*NPPa substrate, a 3 mM stock solution in dimethyl sulfoxide (DMSO) was prepared that was buffered in a 1:1 ratio with a sodium phosphate buffer (0.1 M, pH 6.8) solution. Reaction mixtures containing 50 µL of buffered *p*NPPa and 50 µL of enzyme solution were incubated at 40 °C for 30 min; then, 25 µL of 0.1 M sodium carbonate was added to stop the reaction. The *p*-nitrophenol (*p*NP) release was monitored at 405 nm in 96-well microdilution plates using a SPECTROstar Nano (BMG Labtech, Offenburg, Germany) microplate reader. One enzymatic unit was defined as the amount of the enzyme that releases one µM of *p*NP in one minute under the assay conditions.

### 2.4. Chromogenic Plate Assay

Activity of lipases was evaluated against olive, linseed, peanut, grape seed, rapeseed, almond and menhaden fish oils in plate tests. Chromogenic plates were prepared based on the modified method of Singh et al. [48] and Park et al. [49] to detect the lipase activity of *Chromobacterium viscosum* and *Cordyceps militaris*. The methodology is based on the construction of a solid experimental system in Petri dish containing the corresponding oil substrate and a phenol red compound as indicator. Color of the indicator changes from red to yellow if the pH is lower than 7.4 set for the experiment. Overall, fatty acids are released from the oil by the lipolytic action, which causes a pH decrease; then, a color change can be observed around the disks impregnated with enzyme solution. The level of oil degradation can be evaluated by measuring the width of the yellow zones that formed around the disks in consequence of the hydrolysis of oil substrate. Briefly, 10 mM of calcium chloride solution supplemented with 2% (*w*/*v*) agarose was prepared in distilled water. After mixing using a magnetic stirrer, the pH was adjusted to 7.4 using a 20% (*v*/*v*) sodium hydroxide solution. The mixture was heated in a microwave oven for 3 min until the agarose completely dissolved. The solution was allowed to cool for some minutes and supplemented with 0.01% (*w*/*v*) of phenol red indicator and 1% (*v*/*v*) of the corresponding oil to be tested. This was followed by vigorous shaking to mix both the phenol red and the oil in the solution, from which thin plates were then poured in Petri dishes. Volumes of 10 μL from the 10 mg/mL and 1 mg/mL solutions of *A. niger*, *R. oryzae* and *R. niveus* lipases and from the 1 mg/mL and 0.1 mg/mL of *R. miehei* and *C. rugosa* enzymes were impregnated separately onto filter paper disks (5.0 mm diameter) previously placed on the top of the chromogenic plates. Thus, enzyme activity on the disks was between 13.2 and 21.5 U and between 1.32 and 2.15 U. Heat-inactivated enzyme solutions served as negative controls. Plates were incubated at 30 and 40 °C for 16 h, and the oil degradation was evaluated with the widths of yellow zones (in mm) formed around the disks after 2 h or 16 h. Three independent experiments were performed with each enzyme and natural oil.

### 2.5. Enzymatic Treatment of Oils in a Liquid Environment

In this assay, enzyme-assisted hydrolysis of the tested natural oils was performed in liquid reaction mixtures using the *R. miehei* lipase as the catalyst. A concentration of 4% (*w/v*) of stock solution was prepared from each oil in DMSO, to which phosphate buffer (50 mM, pH 6.8) was added in a 1:1 ratio. The reaction mixture contained 250 µL of buffered oil substrate, 200 µL of phosphate buffer (50 mM, pH 6.8) and 50 µL of 10 mg/mL *R. miehei* lipase solution (1074.7 U). Enzyme-free reaction mixtures were used as controls contained 250 µL of corresponding buffered oil substrate and 250 µL of phosphate buffer (50 mM, pH 6.8). Three independent parallel setups of the reaction mixtures were then incubated for 24 h at 40 °C under constant stirring (200 rpm). After incubation, the mixtures were cooled down to room temperature and stored at −20 °C until GC analysis of fatty acids, and antioxidant and antimicrobial capacity measurements. The reaction conditions (i.e., pH 6.8 and 40 °C) were selected according to a previous study [47].

### 2.6. Extraction of Free Fatty Acids

Free fatty acids from both the enzyme-free (i.e., the control) and the enzyme-containing reaction mixtures were extracted twice with 1 mL hexane, and the combined hexane phases were spiked with 20 µL of 5 mg/mL tridecanoic acid as internal standard and were evaporated to dryness under nitrogen. To methylate the fatty acids, 1 mL of a 14% boron trifluoride–methanol solution (Sigma–Aldrich, Munich, Germany) was added to each sample, and the mixture was incubated at 70 °C for 60 min. After cooling, 0.5 mL of saturated aqueous sodium chloride and 1 mL of hexane were added, and the samples were vigorously mixed by a vortex for 30 s. Finally, the hexane phase was removed, evaporated to dryness under nitrogen and redissolved in 0.5 mL hexane before gas chromatography-mass spectrometry (GC-MS) analysis.

### 2.7. GC-MS Analysis of Fatty Acid Methyl Esters

Fatty acid methyl esters were analyzed by GC-MS on a Shimadzu QP2020 GC-MS equipped with a Shimadzu AOC 6000 autosampler (Shimadzu, Duisburg, Germany). The chromatographic separations were performed using a HP-INNOWax fused silica capillary column (60 m × 0.25 mm i.d., 0.5 µm thickness). Helium was used as carrier gas at a constant flow rate of 1.42 mL/min. The initial GC temperature was 150 °C, after a 2 min hold time was increased at 10 °C/min to 250 °C and then held for 30 min. The injector temperature was kept at 250 °C, and samples were injected in splitless mode (0.5 µL). The transfer line and ion source temperatures were set to 250 and 200 °C, respectively. Full-scan mass spectra were recorded from *m/z* 40 to 400. The mass analyzer was operated in EI (electron impact) ionization mode at 70 eV energy. Mass spectrometry data were acquired and processed with the Shimadzu GCMSsolution version 4.45 software (Shimadzu, Duisburg, Germany). Concentration data of free fatty acids are expressed as µg/mL reaction mixture.

### 2.8. Determination of Antioxidant Capacity

Reactivities toward Folin–Ciocalteu’s (FC) and ferric reducing antioxidant power (FRAP) reagents were studied to analyze the antioxidant properties of samples before and after lipase treatments. The FC reagent is commonly used to assess the phenolic contents of extracts, and the FRAP assay tests the antioxidant capacities of biological samples. The FC reagent can also react with nonphenolic organic substances, such as different aromatic compounds, sugars, proteins or fatty acids [50,51].

In the Folin–Ciocalteu’s reagent reactivity (FCR) assay, 225 µL reaction mixtures contained 30 μL 96% (*v*/*v*) ethanol, 150 μL distilled water, 15 μL 50% (*w*/*v*) FC reagent (Sigma–Aldrich, Munich, Germany) and 30 μL enzyme treated or control samples were prepared. After an incubation at room temperature for 5 min, the reaction was initiated by adding 30 μL 5% (*w*/*v*) sodium carbonate to each mixture. Then, the samples were left in the dark for 60 min, and absorbance was measured at 725 nm using a SPECTROstar Nano spectrophotometer (BMG Labtech, Offenburg, Germany). Oil samples without lipase treatment served as negative controls. FCR of samples is expressed as milligrams of gallic acid equivalent (GAE) in 1 g oil. For the calculations, the standard curve determined from the absorbance data of gallic acid (Sigma–Aldrich, Munich, Germany) solutions with a concentration of 0–100 mg/mL was used.

For the FRAP analysis, the reagent solution prepared contained 80 mL of 300 mM acetate buffer (pH 3.6), 8 mL of 10 mM 2,4,6-tri(2-pyridyl)-s-triazine (TPTZ; Sigma–Aldrich, Munich, Germany) diluted in 40 mM HCl, 8 mL of 20 mM iron(III) chloride and 4.8 mL of distilled water. A volume of 200 µL of FRAP reagent was mixed with 6 µL of enzyme treated or control sample; then, the reaction mixture was left to stand at 37 °C for 30 min. After the incubation, absorbance was measured at 593 nm (SPECTROstar Nano spectrophotometer, BMG Labtech, Offenburg, Germany). Calibration was performed using 1 mM iron(II) sulfate solution in the concentration range of 0.1–1.0 mM. In this assay, antioxidant capacity of samples was expressed as µM Fe(II)/g oil.

### 2.9. Antimicrobial Activity Assays

The effect of hydrolyzed and enzyme-free oils and the minimum inhibitory concentrations (MICs) of selected individual fatty acids on the growth of food-contaminating bacteria, i.e., *Bacillus subtilis* SZMC 0209, *Escherichia coli* SZMC 0582, *Pseudomonas putida* SZMC 6010 and *Staphylococcus aureus* SZMC 0579, were determined by broth microdilution assays. The bacterial strains were provided by the Szeged Microbiology Collection (SZMC, Szeged, Hungary; http://szmc.hu/, accessed on 25 March 2022). To prepare fresh bacterial cultures for the experiments, an inoculum from 24 h plate cultures was transferred into 100 mL Erlenmeyer flasks containing 30 mL lysogeny broth (LB; 10 g/L tryptone, 5 g/L yeast extract, 5 g/L NaCl). The cultures were then incubated for 18 h at 30 °C (in case of *B. subtilis* and *P. putida*) or 37 °C (in case of *E. coli* and *S. aureus*) under continuous shaking (150 rpm). At the end of the incubation period, the growth of each bacterium was in the stationary phase. After incubation, the cultures were diluted to 10-fold with LB medium, and the cell number was determined by Bürker chamber under a light microscope. Stock cell suspensions with a concentration of 10^5^ cells/mL were then prepared from the fresh bacterial cultures in 1× or 2× LB broth depending on the requirements of the applied antimicrobial assay.

To study the antimicrobial effects of hydrolyzed and enzyme-free oils, a volume of 100 μL from the stock cell suspension was transferred to each well of a sterile 96-well microtiter plate (Sarstedt, Nümbrecht, Germany), followed by adding 25 μL from the reaction mixtures previously subjected to boiling for 5 min. Then, the growth environment in each well was adjusted to a 200 μL final volume with 1.25× concentrated LB medium. To determine the MICs of fatty acids, diluted solutions were prepared in buffered (phosphate buffer, 50 mM, pH 6.8) DMSO to achieve a fatty acid concentration range from 62.5 µg/mL to 2 mg/mL. Then, a volume of 100 µL from each fatty acid solution was mixed with 100 µL of stock cell suspension (10^5^ cells/mL) prepared in 2× LB medium. The content of fatty acids in wells thus depicted a range from 31.25 µg/mL to 1 mg/mL. Since the DMSO in reaction mixtures may affect the activity of bacteria tested, a growth control sample, i.e., a positive control, contained bacterial suspension, and 3.125% (*v*/*v*) DMSO was used for proper comparison. Samples containing reaction mixtures or fatty acids in sterile growth medium were considered as negative controls. After setting up the growth environments in plates, their optical density (OD) was measured at 620 nm (SPECTROstar Nano spectrophotometer, BMG Labtech, Offenburg, Germany). Subsequently, the plates were incubated for 24 h at 30 or 37 °C, depending on the requirements of the applied bacterium, and the OD was measured again at the end of incubation. Results are expressed as percent growth (%) calculated by using the following equation:Growth (%) = [(A − B)/(C − D)] × 100
where A and B are the OD of the sample after and before incubation, respectively, and C and D are the OD of the positive control after and before incubation, respectively. Percent growth was calculated from averages obtained from at least three independent parallel inoculations. The concentration of the selected fatty acid compound that caused 90% or higher growth inhibition was considered the MIC.

### 2.10. Statistical Analysis

All measurements were performed in at least three independent, parallel experiments, and data are expressed as averages of the replicates ± standard deviations. Significance was calculated by multiple *t*-test with false discovery rate (FDR) (Q = 10%) or one-way analysis of variance (ANOVA), followed by Tukey’s multiple comparison test (*p* < 0.05) in the GraphPad Prism 7 software (GraphPad Software Inc., San Diego, CA, USA). Pearson’s correlation coefficients (Pearson r) were calculated using GraphPad Prism 7 (GraphPad Software Inc., San Diego, CA, USA).

## 3. Results and Discussion

### 3.1. Screening of Oil Hydrolysis on Chromogenic Plates

Action of the *A. niger*, *C. rugosa*, *R. oryzae*, *R. niveus* and *R. miehei* lipases on the tested oils was studied in the activity ranges of 13.2–21.5 U and 1.32–2.15 U using a chromogenic approach. These activity ranges correspond to the application of the enzymes at the concentrations of 10 and 1 mg/mL (in case of *A. niger*, *R. oryzae* and *R. niveus*) or 1 and 0.1 mg/mL (in case of *R. miehei* and *C. rugosa*), respectively. Since the enzymes used can work differently at different reaction conditions, the hydrolysis capacities of the catalysts were compared at temperatures of 30 and 40 °C and at the 2nd and 16th h of incubation. Heat-inactivated forms of each enzyme were also included in the tests as controls.

Table 1 shows the results of oil degradation with respect to level of hydrolysis for each enzyme. Lipases degraded oil substrates to varying degrees, and wide yellow zones were observed, especially when the biocatalysts were applied in the activity range of 13.2–21.5 U. Considerable degradation was observed with the olive, rapeseed, peanut and linseed oils, even after 2 h of incubation, which, however, varied depending on the enzyme applied for hydrolysis.

Most tested lipases degraded well the olive oil substrate in the plate experiments. In fact, olive oil is a common substrate for identifying lipolytic activities in various assays, e.g., in gel diffusion techniques [52], because it is a preferred substrate for lipase enzymes. The highest degradation capacities on olive oil plates were detected for *R. miehei* and *R. oryzae* lipases during incubation at 40 and 30 °C, respectively (Table 1). Both the lower (0.1 mg/mL) and the higher (1 mg/mL) enzyme concentrations proved to be effective in the case of the *R. miehei* lipase at 40 °C. However, no reaction against the olive oil was observed when 1.48 U of *R. niveus* enzyme was applied for the analysis at 30 and 40 °C temperatures. Prolonging incubation time up to 16 h did not cause a considerable change in the olive oil degradation by *R. miehei*, *R. oryzae* or *R. niveus* enzymes. On the contrary, the olive oil hydrolysis by *C. rugosa* and *A. niger* lipases at 30 and 40 °C, respectively, resulted in degradation zones between 2 and 2.9 mm after the 2nd hour of incubation (Table 1), which was more intense (3.8 mm and 3.2 mm zones for the *C. rugosa* and *A. niger* enzymes, respectively) by the 16th hour of incubation (data not shown). The *C. rugosa*, *A. niger* and *R. oryzae* effectively hydrolyzed the rapeseed oil, especially at higher enzyme concentrations. *C. rugosa* and *R. oryzae* lipases worked at 30 °C, while *A. niger* at both temperatures showed yellow zones between 2 mm and 2.9 mm. In addition, *R. niveus* at 40 °C also showed high hydrolyzing capacity against rapeseed oil. However, interestingly, there was no or slight degradation when *R. miehei* lipase was used as the catalyst during the 2-h reaction. Prolonging incubation up to 16 h, however, resulted in 2.2 and 2.7 mm degradation zones at 30 and 40 °C, respectively, for the *R. miehei* lipase on rapeseed oil (data not shown). Recently, hydrolysis studies on rapeseed oil were reported using *A. niger* [45] and *Thermomyces lanuginosus* [53] lipases. In addition, the lipase of *R. miehei* has successfully been applied for transesterification of rapeseed oil with methanol [54].

Application of 21.3 U *C. rugosa* and 18.8 U *R. oryzae* lipases resulted in the highest hydrolysis against peanut oil demonstrating 3 mm degradation zones (Table 1). However, other enzymes tested also showed efficient hydrolysis on peanut-oil-containing plates, even at lower concentrations. Concerning linseed oil, a high degree of hydrolysis was observed for all enzymes tested at both 30 and 40 °C temperatures (Table 1). Chen et al. [43] screened the hydrolysis of linseed oil with four commercially available lipases. The *C. rugosa* (Lipase AY, Amano Enzyme Inc., Nagoya, Japan) enzyme exhibited the best efficiency with a 91.79% hydrolysis ratio at 30 °C. Similarly to our study, the *C. rugosa* lipase was highly active (more than 80% hydrolysis ratio) at 40 °C as well. Efficient linseed oil hydrolysis with degradation zones of 4.2 mm (at 30 °C) and 3.8 mm (at 40 °C) was observed, even after 16 h incubation, when the *A. niger* lipase was used as the catalyst (data not shown). This phenomenon was not detected for the other enzymes tested. Hydrolysis of almond oil was moderate compared to that of other vegetable oils tested during the initial part of incubation. However, a visually well distinguishable degradation zone could be observed for most enzymes after 16 h of incubation (Table 1). On almond oil containing plates, the *R. miehei* and *R. oryzae* lipases worked well at both 30 and 40 °C temperatures, presenting hydrolysis zones between 2 and 3.9 mm. In fact, overall degradation of almond oil was highest when the *R. miehei* enzyme was used as a catalyst. The *C. rugosa* lipase showed the least hydrolyzing ability on almond oil, presenting maximal degradation zones of 1.4 mm during reactions. When grape-seed-oil-containing plates were prepared, a color change from red to yellow was observed after the chromogenic plate solidified. Although the pH of the solid environment was not determined, it may have turned acidic due to the addition of grape seed oil to the reaction medium. The pH-based detection method applied in our experiment is highly sensitive, since the phenol red indicator turns yellow upon slight acidification from the adjusted 7.4 pH [48]. Anyway, due to the yellow background appearing in the whole reaction medium, the colorimetric plate assay could not be applied for grape seed oil.

An oil sample from menhaden fish was also included in the hydrolysis screening studies. This animal oil was digested well by all enzymes tested, especially at the working temperature of 40 °C (Table 1). At the 16th h of incubation, the highest degradation zones were detected for the enzymes from *R. miehei*, *A. niger* and *R. oryzae*. Moreover, when the *R. miehei* lipase was used as the catalyst, the widths of zones around the disk were more than 4.5 mm at both 30 and 40 °C incubation temperatures. The *A. niger* and *R. oryzae* lipases demonstrated hydrolysis zones between 3 and 3.9 mm at 40 °C, which also corresponds to strong degradation (Table 1).

### 3.2. Lipolysis in Liquid Environment

In this assay, liquid reaction mixtures were prepared, and the major fatty acids released during enzymatic oil degradation were analyzed. The *R. miehei* lipase was selected as the catalyst in this experiment because of its efficient hydrolytic property against most oils involved in the chromogenic plate assays. In addition, *R. miehei* lipase is an industrially important enzyme with frequent utilization in many biotechnological processes [55], including production of special fatty acids from lipids, alcoholysis reactions and esterification of various fatty acids [56]. Moreover, *R. miehei* lipase can be produced in high yields by fermentation approaches [57], and stability of the produced enzyme can be maintained even under harsh reaction conditions. Fatty acids released during *R. miehei* lipase treatment of oils were monitored by GC-MS technique. Then, the free fatty acid content of each treated sample was compared to that of the corresponding unhydrolyzed control mixture. As shown in Figure 1 and Table 2, concentrations of major fatty acids identified in the reaction mixtures mostly increased during the enzyme treatment.

In vegetable-oil-containing mixtures, in general, concentrations of the saturated free fatty acids, i.e., PA and SA, and the unsaturated free compounds of OA, LA and ALA, showed marked increases after the enzyme treatments. The relative contents of free PA, SA, OA, LA and ALA fatty acids were 10.8–71.1%, 5.8–18.1%, 3.5–36.1%, 10.1–33.9% and 0–60.1% in hydrolysates, respectively, depending on the oil used. For olive oil, concentrations of free OA, LA, PA and ALA were considerably increased by 8.6-, 2.8-, 1.8- and 1.4-fold, respectively, at the end of the lipolytic treatment (*p* < 0.05) (Figure 1A). Enzymatic hydrolysis of olive oil was also effectively conducted in the study of Ferreira et al. [58], using a crude lipase extract from *Geotrichum candidum*. They found OA and LA production, but free ALA was not identified during the hydrolysis. The differences in the contents of major free fatty acids after lipase treatments can be explained by the different substrate specificities of the enzymes used and/or the different sources of oils subjected to hydrolysis. Increases by 6.9- and 2.5-fold (*p* < 0.01) were detected for free OA and LA, respectively, in the almond oil sample after the hydrolytic treatment (Figure 1B). Additional free unsaturated fatty acids could not be detected in either enzyme-free or treated almond oil samples. LA release was significant (*p* < 0.05) during the peanut oil degradation as well (Figure 1C). In addition, free PA content in peanut oil hydrolysate was also considerable, in contrast to the findings documented by Soumanou et al. [59] during lipase-assisted hydrolysis with free and immobilized varieties of the *R. miehei* lipase. For rapeseed and linseed oils, results revealed a conspicuous change in the free fatty acid content after hydrolysis. In rapeseed oil, for instance, the quantities of unsaturated free fatty acids, i.e., OA, LA and ALA, were elevated significantly during enzyme treatment (*p* < 0.05); and there was no remarkable increase in SA or PA content (Figure 1D). This may have been due to the fact that OA, LA and ALA are major fatty acids in rapeseed oil triglycerides, whereas SA and PA can be found in moderate percentages [45]. Concerning linseed oil, both the saturated and unsaturated free fatty acid contents increased dramatically (*p* < 0.05); however, unsaturated fatty acid molecules were released in a higher proportion than saturated ones (Figure 1E). In particular, the 2.19 µg/mL initial free ALA content for the linseed oil-containing sample increased remarkably to 84.26 µg/mL, a 38.3-fold increase (*p* < 0.0001) (Figure 1E). Although the *R. miehei* lipase was characterized by a 1,3-regiospecific nature, ALA was the dominant fatty acid in linseed oil hydrolysate, as in the results of Chen et al. [43], obtained after lipolysis with the non-regioselective *C. rugosa* lipase (Lipase AY). With this context, non-positional specific lipases resulted in higher rates of linseed oil hydrolysis, together with higher concentrations of free ALA, than the 1,3-specific ones in the study of Rupani et al. [60]. For grape seed oil samples, a moderate increase (*p* > 0.05) in the amount of free unsaturated fatty acids was recorded during the lipolysis (Figure 1F). However, it was shown that the major fatty acids found in grape seed oil triglycerides [7,36], i.e., LA and OA, can be released by the *R. miehei* lipase treatment conducted. In addition, it is worth mentioning the significant (*p* < 0.0001) improvement in free PA content of the grape seed oil sample after hydrolysis (Figure 1F).

Hydrolysis of menhaden fish oil by microbial lipases has been frequently studied in the last few decades [61,62,63,64,65]. Most of these examinations were mainly focused on the production of EPA and DHA-enriched mixtures by using free and/or immobilized enzyme preparations. This work revealed the release of additional fatty acids, including a variety of PUFAs, during an *R. miehei* lipase mediated reaction. An analytical assay revealed increases in the concentrations of sixteen fatty acid types present in free form after hydrolysis (Table 2). The concentrations of saturated—i.e., PA, SA and arachidic acid; and monounsaturated—i.e., PLA, OA and eicosenoic acid—free fatty acids were increased by approximately 2- or 3-fold by the enzyme treatment, resulting in final relative contents of 33.7, 7.1, 0.6, 27.5, 5.3 and 1.9%, respectively, in the hydrolysates. Significant (*p* < 0.05) liberation for PUFAs, such as hexadecanedioic acid (HDA), LA, ALA, stearidonic acid, arachidonic acid, EPA, eicosadienoic acid, docosapentaenoic acid, tetracosenoic acid and DHA, was detected after the lipase hydrolysis (Table 2); relative contents were 4.1, 2.1, 3.8, 1.1, 0.9, 6.4, 0.7, 1.4, 1.1 and 2.2%, respectively. Of these molecules, the latter four free fatty acids were not detectable in the enzyme-free sample. Considerable increases, i.e., more than 3.5-fold, were recorded for free HDA, LA, ALA and EPA contents during the hydrolysis process compared to the unhydrolyzed control. Free DHA content in the hydrolysates was moderate compared to EPA. This may be attributed to the preference of *R. miehei* lipase for EPA in menhaden oil, as it has been documented in the study of Mohammadi et al. [64]. Additionally, Fernández-Lorente et al. [66] investigated the sardine oil hydrolysis by immobilized *R. miehei* lipase and found that the release of EPA was faster than that of the DHA due to the nature of the selectivity of the enzyme. EPA in menhaden oil is mostly located at *sn*-2 and *sn*-3 positions [61], of which 1,3-specific lipases, such as the *R. miehei* enzyme used in this study, are able to release the compound from the *sn*-3 position. The 1,3-specific nature of *R. miehei* lipase supported the release of ALA and tetracosenoic acid (C24:1) compounds as well, which can be found at the *sn*-1,3 position in menhaden oil [67]. However, most DHA fatty acids in menhaden fish oil are on the *sn*-2 position of the triglyceride molecules [67], which may explain the moderate free DHA content after hydrolysis.

### 3.3. Bioactive Properties of the Hydrolysates

#### 3.3.1. Antioxidant Capacity

As the *R. miehei* lipase treatment had a positive effect on the concentrations of some fatty acids in the oils tested, and studies reported release of phenolic compounds from esters by lipolytic activities [68], phenolic content and antioxidant potential were screened before and after the hydrolysis by analyzing the FCR and the FRAP capacities, respectively.

Results showed that the lipase treatment positively affected both the FCR and the antioxidant activity of the natural oils involved. The FCR showed 1.9- to 4.3-fold increases in the enzyme treated samples compared to the unhydrolyzed control (Figure 2A). The highest increment in FCR was found for the rapeseed (*p* < 0.0001) and grape seed (*p* < 0.01) oil samples after the hydrolysis, but the other hydrolyzed oils showed also significantly enhanced FCR during the reaction. Except olive oil, the FRAP of the samples also exhibited considerable increases (at least *p* < 0.05) after incubation with the *R. miehei* lipase (Figure 2B). The overall FRAP of linseed and grape seed oils improved most significantly, presenting about 1.6 (*p* < 0.01) and 1.8 (*p* < 0.0001) times increases, respectively, in the lipolytic surrounding compared to the enzyme-free control. The increases in FCR and FRAP capacity may suggest *R. miehei* lipase-supported release of phenolic compounds from their esters (e.g., from phenolic triglycerides and other lipophenols) found in plant materials. In the case of menhaden oil, FCR and FRAP reactivity could be attributed to the presence of some vitamin [16] degradation products, for instance, the retinoic acid that can react with the FC and FRAP reagents with a good affinity [51,69]. However, further experiments, e.g., identification and determination of individual phenolics in samples, are needed to prove the above assumptions.

Considering fatty acids identified in samples, correlation analysis revealed positive associations among FCR (*r* = 0.388) and FRAP activity (*r* = 0.604) and PA concentration of lipase treated plant oils. Similarly, a direct relationship between PA and total antioxidant capacity has been documented by Szabó et al. [29] in grape pomace samples. Additionally, they reported that the amounts of ALA and LA correlated well with the antioxidant activity of the pomaces. During the present research, no and moderate associations between FCR and ALA (*r* = −0.039) and LA (*r* = 0.684) concentrations were observed. However, the LA content of hydrolysates correlated well (*r* = 0.928, *p* < 0.01) with their FRAP activity. Additionally, SA and OA concentrations of treated plant oils were found to moderately correlate (*r* = 0.448 and *r* = 0.403, respectively) with the FRAP potential measured. A slight correlation has been described between the SA content and the antioxidant activity in avocado samples as well [70]. De Alencar et al. [28] did not identify any dose-dependent relationship between fatty acid content and FRAP activity in extracts of algae samples; however, ferrous ions chelating activity and β-carotene bleaching assays performed by them showed an increased antioxidant activity for samples containing more saturated fatty acids than unsaturated. In the study of Henry et al. [26], the saturated and unsaturated fatty acids showed different antioxidant activities, which were also affected by the carbon chain length of the molecules. Namely, high antioxidant activity was verified for medium- and long-chain saturated fatty acids, such as lauric, myristic and palmitic acids, and for most unsaturated fatty acids tested [26]. In addition, a clear association between the unsaturated fatty acid content and antioxidant potential has been demonstrated by Huang and Wang [71] and Karaman et al. [72] in seaweed and mushroom samples, respectively. Nevertheless, our work revealed a potential relationship of antioxidant capacity with free fatty acids present in vegetable and fish oils hydrolysates, which, to the best of our knowledge, has not been reported so far.

#### 3.3.2. Antimicrobial Activity

Many individual fatty acids released during the *R. miehei* lipase treatment have shown antimicrobial potential according to previous reports. Hence, the next step in this study was to investigate the growth inhibitory effects of the prepared hydrolysates against food-contaminating microorganisms, i.e., *B. subtilis*, *E. coli*, *P. putida* and *S. aureus*. In this assay, the antimicrobial activity of the hydrolysates was tested at a lipid concentration of 1.25 mg/mL in a microdilution system. The bacterial growth was compared to that obtained in the sample with the same amount of unhydrolyzed materials. To consider the potential growth inhibitory effect of DMSO present in the hydrolysates, a lipid-free growth control sample contained DMSO at the corresponding concentration was also included in the assay. As a positive control, the bacterial growth in this sample was taken as 100%. DMSO can affect the growth of many microorganisms in a concentration-dependent manner; however, studies have shown its favorable use as a solvent in antimicrobial tests [73,74]. In *E. coli*, for instance, an MIC of 15% (*v*/*v*) was determined for DMSO [75]. Bacteria included in our study grew at the DMSO concentration of 3.125% (*v*/*v*) present in the reaction mixtures.

An inhibitory effect was observed for many samples containing hydrolyzed or unhydrolyzed oils against the growth of bacteria tested (Table 3). Overall, the lipase-treated linseed, grape seed and menhaden fish oils significantly (*p* < 0.05) inhibited the growth of all bacteria compared to the positive control. The hydrolyzed olive oil showed an inhibitory effect only against *S. aureus*, whereas the enzyme-treated peanut oil decreased the *B. subtilis* and *S. aureus* growth (Table 3). For some oils tested, however, the untreated sample already showed an inhibitory effect compared to positive control, which varied depending on the bacterium investigated. For instance, enzyme-free samples of linseed, grape seed and menhaden fish oils were also able to significantly inhibit the activity of both the *E. coli* and the *S. aureus* (*p* < 0.05). *S. aureus* growth was also decreased in the presence of unhydrolyzed olive and peanut oils. A significant inhibitory effect was identified for the enzyme-free grape seed oil sample on *B. subtilis* as well (*p* < 0.05) (Table 3). Depending on the bacterium studied, however, the lipase treatment altered the growth-affecting properties of some oils. Increased antimicrobial activity for enzyme-treated samples of linseed oil, grape seed oil and menhaden fish oil was observed against *B. subtilis* and *P. putida* compared to their untreated activity (Table 3). Hydrolysates of peanut oil and menhaden fish oil also demonstrated improved activity against *B. subtilis* and *S. aureus*, respectively, after the hydrolysis. The antimicrobial activity-promoting effect of *R. miehei* lipase treatment has been described for other plant oils as well [76]. Among the seed oils investigated in that study, hydrolyzed samples of bitter gourd and lady’s finger seed oils proved to be the best antimicrobials against the tested food-contaminating bacteria, i.e., *Salmonella typhimurium*, *Listeria monocytogenes* and *E. coli*. Lipase hydrolyzed plant oils exhibited action towards *Clostridium perfringens*, *Enterococcus cecorum*, *L. monocytogenes* and *S. aureus* (Gram-positive bacteria) at 0.5–4.5 mg/mL in the study of Hovorková et al. [77]. Concerning almond and rapeseed oil hydrolysates, there was no effect on the growth of any bacteria compared to the positive controls. In contrast, enzyme-free samples of them caused significant inhibition of *E. coli* and *S. aureus* growth. A similar result was registered for the peanut oil and olive oil against *E. coli* and *P. putida*, respectively, when the effects of untreated and treated samples were compared (Table 3).

Regarding the bacteria involved in this study, growth of both *B. subtilis* and *P. putida* was tolerant of most unhydrolyzed oils tested; however, hydrolysates from linseed, grape seed and menhaden fish oils resulted in a very strong inhibitory effect against them. It is worth mentioning that most enzyme-free samples had a positive effect on the growth of *B. subtilis* compared to the positive control (Table 3). The growth-promoting effect of lipid materials against bacteria was supported by other reports as well [78,79]. *E. coli* and *S. aureus* have been shown to be sensitive to most oils and their hydrolysates. Among the analyzed oils, samples of linseed and grape seed oils and the hydrolyzed mixture of menhaden oil exhibited the highest activity against the growth of the bacteria tested (Table 3). Although the antimicrobial properties of essential oils is an intensively studied topic, there are a few available data on the effects of hydrolyzed and unhydrolyzed vegetable and fish oil samples in the literature. The antimicrobial effects of grape seed and olive oils have been widely studied against food-related microorganisms. For instance, their inhibitory effect on *S. aureus* and *E. coli* growth has been published [7,80,81]. In the study of Dabetic et al. [82], grape seed oil was also a high-activity agent against *S. aureus* but had no inhibition towards *B. subtilis* and *E. coli*. Growth of Gram-positive and Gram-negative bacteria, i.e., *Bacillus cereus*, *L. monocytogenes*, *S. aureus*, *Pseudomonas aeruginosa*, *Yersinia enterocolitica*, *S. typhimurium* and *E. coli*, has shown to be efficiently inhibited by linseed and fish oil preparations in a recent study [18]. In addition, fish oil proved to be effective inhibitor towards the growth of *E. coli* and *S. aureus* in the study of Silva et al. [83]. In contrast to our results obtained via broth microdilution tests, a recent publication for peanut oil described inactivity against both *E. coli* and *S. aureus* in an agar diffusion assay [84].

Individual fatty acids may possess antimicrobial properties against food-contaminating microorganisms. In this context, a correlation analysis was performed to evaluate relationships between the antimicrobial effects and individual fatty acid contents of oil samples after lipase-treatment. As can be seen in Table 4, strong positive correlations (*r* > 0.911, *p* < 0.01) between the amount of LA and the antimicrobial capacities of treated sample toward *B. subtilis*, *E. coli* and *P. putida* were identified. The association proved to be moderate (*r* = 0.757, *p* < 0.05) when *S. aureus* was used as the testing microorganism. ALA concentration in the hydrolyzed oil samples was also highly correlated (*r* = 0.806, *p* < 0.05) with the influence on *P. putida* growth. In addition, ALA content had moderate relationships (*p* > 0.05) with the antimicrobial effects of samples against *B. subtilis* and *E. coli* (Table 4). Studies have shown that omega-3 PUFAs can be excellent antimicrobial agents [1,85,86]. For most oil samples involved in our study, the major omega-3 fatty acids released by *R. miehei* lipase treatment were ALA and EPA. These compounds, as seen in the case of ALA (Table 4), can be responsible for altering the antimicrobial effects of treated oil samples. Hence, it was considered important to assay their MICs against the tested food contaminants within the range of 1 mg/mL to 31.25 µg/mL. The broth microdilution assay revealed resistance of *E. coli* and *S. aureus*, even at the highest ALA and EPA concentrations tested (Table 5). For comparison, a value of 256 µg/mL of MIC_50_ was identified for EPA against different *S. aureus* strains involved in the study of Desbois and Lawlor [74], indicating variable susceptibility of different strains towards the inhibitor. In our study, the most susceptible bacterium towards ALA and EPA was *B. subtilis*, since both compounds resulted in an MIC within the tested range. Similarly, *B. subtilis* was among the bacteria sensitive towards EPA in the work of Shin et al. [22]. The MIC value of 350 µg/mL presented concerned bioconverted EPA; however, interestingly, the non-bioconverted EPA compound exhibited at most a weak antimicrobial effect in contrast to our results. An MIC value within the tested range was found for EPA in *P. putida* as well (Table 5).

In view of the above results, it seems that LA, ALA and/or EPA contents of the corresponding hydrolyzed samples may be responsible for the antimicrobial activity identified against *B. subtilis* and *P. putida*. However, as can be clearly deduced from the results obtained for *E. coli* and *S. aureus* as well, the presence of other bioactive compounds, i.e., fatty acids and/or phenolics, in the hydrolysates, and a possible synergic effect [83] between these molecules, could also affect the antimicrobial potential of the samples. For instance, PA proved to be an efficient inhibitor, even at a concentration of 100 µg/mL, against *E. coli* in the study of Padmini et al. [87]. The mechanism of action of the free fatty acids against bacteria can be ascribed to several factors, including disruption of the cytoplasmic membrane; inhibition of DNA/RNA replication and cell wall and protein synthesis routes; and alterations to enzyme activities and metabolic processes [21]. Several studies have reported more potent activity by various medium and long chain fatty acids on Gram-positive bacteria than on Gram-negative ones [85,88]. The less potent activity against Gram-negative bacteria may be due to the protective effect of their outer membrane against hydrophobic substances [89]. In our studies, however, growth of Gram-positive and Gram-negative bacteria reacted differently to each sample, depending on the oil and hydrolysate. For instance, of the bacteria included in the assay, the Gram-positive *B. subtilis* and the Gram-negative *E. coli* showed the highest sensitivity to linseed, grape seed and menhaden fish oil samples, which proved to be as the most potent inhibitors.

## 4. Conclusions

Microbial lipases are important biocatalysts with which to modify oil substances used in the food industry. Although there are studies on the ability of lipases to release fatty acids in some edible oils, the bioactive properties of the produced hydrolysis products have not yet been characterized as far as we know. In this work, firstly, the hydrolysis capacities of fungal lipases, i.e., commercial enzymes from *A. niger*, *C. rugosa*, *R. oryzae*, *R. niveus* and *R. miehei*, were screened in a chromogenic plate test containing olive, rapeseed, linseed, almond, peanut, grape seed or menhaden fish oils as the substrates. Olive, rapeseed, peanut and linseed oils showed a high degree of degradation, and results revealed the highest activities for lipases obtained from *R. miehei*, *A. niger* and *R. oryzae*. Next, the oil hydrolysis was analyzed in a liquid environment optimized for the *R. miehei* lipase (selected as a catalyst). Chromatography showed marked increases in the concentrations of the saturated PA and SA, and the unsaturated OA, LA and ALA fatty acids from most vegetable oils after the lipolytic treatment. In addition, other PUFAs, e.g., EPA and DHA, were also liberated in considerable amount from the menhaden fish oil during the hydrolysis. The *R. miehei* lipase treatment improved both the Folin reactivity and the reducing power of many oils involved. In antimicrobial activity assays, the linseed, grape seed and menhaden fish oil samples proved to be potent antimicrobial agents against food-contaminating bacteria involved. Moreover, the lipase treatments had a stimulative effect on the antimicrobial activity of several oils tested. In conclusion, the *R. miehei* lipase treatment may be a suitable approach to developing bioactive lipid mixtures with antioxidative and antimicrobial activities from vegetable and fish oil substances. The health-protective fatty acid compounds obtained after an extraction can be used as additives in functional food products.

## Figures and Tables

**Figure 1 foods-11-01711-f001:**
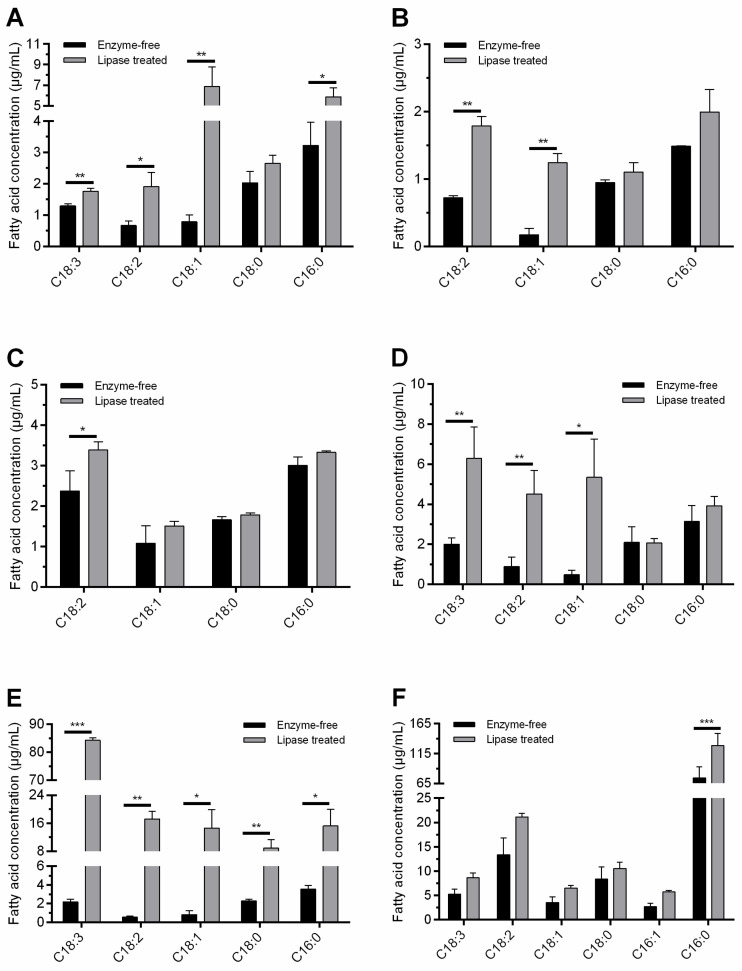
Concentration of major free fatty acids (µg/mL reaction mixture) determined by GC-MS in enzyme-free and *R. miehei* lipase treated olive (**A**), almond (**B**), peanut (**C**), rapeseed (**D**), linseed (**E**) and grape seed (**F**) vegetable oil substances. Lipolytic reactions were performed at 40 °C for 24 h. Fatty acids: α-linolenic acid, C18:3; linoleic acid, C18:2; oleic acid, C18:1; stearic acid, C18:0; palmitoleic acid, C16:1; palmitic acid, C16:0. Results presented are averages of concentration values determined in three replicates; error bars represent standard deviations. Asterisks indicate significant differences between the enzyme-free and treated samples according to multiple *t*-tests performed by GraphPad Prism version 7.00, FDR (Q = 10%), * *p* < 0.05, ** *p* < 0.01, *** *p* < 0.0001.

**Figure 2 foods-11-01711-f002:**
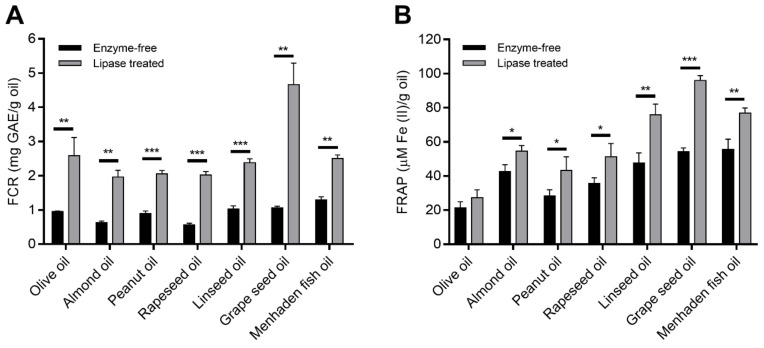
Comparison of FCR (**A**) and FRAP activity (**B**) of enzyme-free and *R. miehei* lipase-treated olive, almond, peanut, rapeseed, linseed, grape seed and menhaden fish oil samples. Results are means of data of three replicates; error bars represent standard deviations. Asterisks indicate significant difference between the enzyme-free and treated samples according to a multiple *t*-test performed in GraphPad Prism version 7.00, FDR (Q = 10%), * *p* < 0.05, ** *p* < 0.01, *** *p* < 0.0001.

**Table 1 foods-11-01711-t001:** Hydrolysis of natural oils by commercial lipases on chromogenic plates incubated at 30 °C and 40 °C temperatures. The width of the yellow color zone is proportional with the level of hydrolysis. The data presented show the best degradation capacity of each enzyme for a given oil.

Lipases	Activity (U)	Oil Hydrolysis at 30 °C and 40 °C Temperatures ^1^											
		Olive Oil		Almond Oil		Rapeseed Oil		Peanut Oil		Linseed Oil		Menhaden Oil	
		30	40	30	40	30	40	30	40	30	40	30	40
*C. rugosa*	21.3	+++	++	++	++	+++	++	++++	+++	+++	+++	++	+++
	2.13	++	ND	ND	ND	++	++	+++	+++	+++	+++	ND	+++
*R. miehei*	21.5	++	++++	++++	++++	ND	+	+++	+++	+++	+++	+++++	+++++
	2.15	+	+++	++	+++	ND	ND	+++	+++	+++	+++	+++	+++
*A. niger*	13.2	++	+++	++	+++	+++	+++	+++	++	+++	+++	+++	++++
	1.32	+	++	+	ND	+	++	+++	++	ND	+++	++	++
*R. oryzae*	18.8	++++	+++	+++	+++	+++	++	+++	++++	+++	+++	+++	++++
	1.88	+	+	++	+++	+	+	++	+++	++	++	++	+++
*R. niveus*	14.8	+++	++	+++	+	+	+++	+++	+++	+++	+++	+++	+++
	1.48	ND	ND	+++	ND	ND	++	ND	++	++	++	++	++

^1^ Slight degradation (+), yellow zone: 0.5–0.9 mm; moderate degradation (++), yellow zone: 1.0–1.9 mm; high degradation (+++), yellow zone: 2.0–2.9 mm; strong degradation (++++), yellow zone: 3.0–3.9 mm; very strong degradation (+++++), yellow zone: >4.0 mm; ND: no degradation. Presented results were obtained after the 2nd hour of incubation for olive, rapeseed, peanut and linseed oils, and on the 16th hour of incubation for almond and menhaden fish oils.

**Table 2 foods-11-01711-t002:** Concentrations of free fatty acids detected in menhaden fish-oil-containing mixtures by GC-MS before (enzyme-free control) and after treatment with *R. miehei* lipase (40 °C, 24 h).

Fatty Acids	Fatty Acid Concentration (µg/mL Reaction Mixture)	
	Enzyme-Free Control	*R. miehei* Lipase Treated
Palmitic acid (C16:0)	141.31 ± 7.74	278.38 ± 87.68
Palmitoleic acid (C16:1)	111.96 ± 6.06	227.01 ± 61.75
Hexadecanedioic acid (C16:2)	7.65 ± 1.45	33.28 ± 4.77 **
Stearic acid (C18:0)	29.51 ± 1.73	58.96 ± 18.22
OA (C18:1)	14.58 ± 2.08	43.81 ± 12.61 *
LA (C18:2)	4.67 ± 0.32	16.93 ± 4.54 *
ALA (C18:3)	9.02 ± 0.57	31.58 ± 7.01 *
Stearidonic acid (C18:4)	4.32 ± 0.07	8.52 ± 0.99 **
Arachidic acid (C20:0)	2.31 ± 0.65	4.55 ± 1.16
Eicosenoic acid (C20:1)	9.27 ± 0.39	16.27 ± 3.57
Eicosadienoic acid (C20:2)	n. d. ^1^	5.95 ± 0.17 ***
Arachidonic acid (C20:4)	5.41 ± 0.14	7.49 ± 0.49 **
EPA (C20:5)	13.74 ± 1.09	52.85 ± 11.5 *
Docosapentaenoic acid (C22:5)	n. d.	12.02 ± 0.68 ***
DHA (C22:6)	n. d.	18.7 ± 2.91 **
Tetracosenoic acid (C24:1)	n. d.	8.49 ± 1.16 **

^1^ Not detected. Asterisks indicate significant differences between the enzyme-free and treated samples according to a multiple *t*-test performed by GraphPad Prism version 7.00, FDR (Q = 10%), * *p* < 0.05, ** *p* < 0.01, *** *p* < 0.0001.

**Table 3 foods-11-01711-t003:** Effects of the olive, almond, peanut, rapeseed, linseed, grape seed and menhaden fish oil samples (1.25 mg/mL lipid concentration) on growth of food-contaminating bacteria before and after treatment with *R. miehei* lipase. Growth in lipid-free environment containing 3.125% (*v/v*) DMSO was taken as 100% (positive control).

Oil Materials		Growth (%) ^1^			
		*B. subtilis*	*E. coli*	*P. putida*	*S. aureus*
Positive Control		100 ± 0 a	100 ± 0 a	100 ± 0 a	100 ± 0 a
Olive oil					
	Enzyme-free	120.8 ± 8.1 bc	72.4 ± 6.9 abc	75.9 ± 7.6 bc	59.5 ± 5.4 bc
	Lipase treated	125.1 ± 5.2 bd	83.5 ± 14.7 ab	88.7 ± 8.4 ab	78.7 ± 4.3 def
Almond oil					
	Enzyme-free	113.1 ± 6.9 be	65.9 ± 13.8 bcd	101.6 ± 12.1 a	57.3 ± 5.7 bc
	Lipase treated	103.6 ± 4.4 ae	94.5 ± 3.7 ad	94.1 ± 3.5 a	88.3 ± 7.1 ad
Peanut oil					
	Enzyme-free	106.5 ± 7.8 ae	60.8 ± 7.8 be	101.4 ± 2.5 a	44.9 ± 2.5 b
	Lipase treated	80.9 ± 6.3 f	72.9 ± 12.1 ade	89.3 ± 7.1 ab	72.7 ± 3.8 cdfg
Rapeseed oil					
	Enzyme-free	135.8 ± 0.6 dg	57.6 ± 1.2 bce	100.3 ± 0.4 a	61.2 ± 5.7 bf
	Lipase treated	128.5 ± 1.2 cd	97.2 ± 15.1 ad	86.1 ± 1.9 abd	93.8 ± 7.1 ae
Linseed oil					
	Enzyme-free	113.3 ± 2.2 be	50.5 ± 7.8 ce	95.3 ± 11.2 a	55.1 ± 1.7 bg
	Lipase treated	0.0 ± 0.0 h	56.1 ± 11.2 bce	48.8 ± 1.4 e	74.5 ± 7.2 cdf
Grape seed oil					
	Enzyme-free	45.7 ± 1.5 i	43.8 ± 10.1 ce	91.8 ± 6.3 ab	50.3 ± 9.2 b
	Lipase treated	0.0 ± 0.0 h	47.1 ± 10.6 ce	58.6 ± 3.2 ce	54.8 ± 3.1 bg
Menhaden fish oil					
	Enzyme-free	147.9 ± 2.2 g	52.6 ± 6.7 bce	83.8 ± 3.5 abd	100.3 ± 11.9 a
	Lipase treated	0.0 ± 0.0 h	52.3 ± 18.2 bce	69.2 ± 2.6 cd	69.2 ± 0.7 cfg

^1^ Values are averages computed from three tests ± standard deviations. Values within a column with different letters are significantly different according to the one-way ANOVA followed by Tukey’s multiple comparison test (*p* < 0.05).

**Table 4 foods-11-01711-t004:** Correlation coefficients (Pearson r) between the antimicrobial activity against different food-contaminants and palmitic acid (PA), stearic acid (SA), oleic acid (OA), linoleic acid (LA) and α-linolenic acid (ALA) contents of hydrolyzed oil samples.

Fatty Acids	*B. subtilis*	*E. coli*	*P. putida*	*S. aureus*
PA (C16:0)	0.686	0.667	0.405	0.559
SA (C18:0)	0.613	0.562	0.345	0.364
OA (C18:1)	0.625	0.547	0.416	0.291
LA (C18:2)	0.976 **	0.911 **	0.931 **	0.757 *
ALA (C18:3)	0.674	0.523	0.806 *	0.171

Asterisks indicate significant correlations according to correlation analysis performed by GraphPad Prism version 7.00, * *p* < 0.05, ** *p* < 0.01.

**Table 5 foods-11-01711-t005:** Minimum inhibitory concentrations (MICs) of α-linolenic acid (ALA) and eicosapentaenoic acid (EPA) fatty acids against food-contaminating bacteria.

Fatty Acids	MIC (µg/mL)			
	*B. subtilis*	*E. coli*	*P. putida*	*S. aureus*
ALA (C18:3)	125	>1000	>1000	>1000
EPA (C20:5)	62.5	>1000	500	>1000

## Data Availability

Data are contained within the article.

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
