# Peer review of "Hydrolysis of Edible Oils by Fungal Lipases: An Effective Tool to Produce Bioactive Extracts with Antioxidant and Antimicrobial Potential"

_foods, 2022, doi:10.3390/foods11121711_

Round 1
Reviewer 1 Report
The authors have presented and interesting study regarding the enzymatic hydrolysis with lipases of oils from different origins and the characterization of deriving products. Therefore, in the discussion they focused on the presence of specific free fatty acids and attributed the antioxidant capacity and antimicrobial activity of the hydrolyzed oils to the presence of specific free fatty acids. Moreover, they only briefly commented the presence of phenolic compounds in the oils (lines 444-447). I believe that this aspect need to be discussed a little more as vegetal oils are rich in phytochemicals and phenolic compounds that are responsible for antioxidant and antimicrobial activity.
The authors verified whether the enzymatic de-esterification of triglycerides, resulting in the release of free fatty acids, can also cause a weakening of the hydrophobic interactions between the triglycerides themselves and the phenolic compounds, favoring the mobility of the latter? This could explain the improvement in antioxidant and antimicrobial properties attributed, instead, to the phytochemical compounds and not to the fatty acids themselves.
Other comments:
Line 109- I would suggest to change the abbreviation pNPP for p-Nitro Phenyl Palmitate in pNPPa as pNPP is usually used for p-Nitro Phenyl Phosphate and could be confusing
Line 183 – “were studied” instead of “was studied”
Line 184 – FC reagent is used to assess the phenolic content and not the antioxidant capacity, please reformulate the sentence.
Reviewer 2 Report
The manuscript “Hydrolysis of vegetable and fish oils by fungal lipases: an effective tool to produce bioactive cocktails with antioxidant and antimicrobial potentials” is generally very well written and contains data of some relevance for a general readers as well as of high relevance for specialists in the topic. Although the subject of the paper could be of interest for the readers of the journal, the paper needs some corrections:
- Page 2, lines: 46,50,53; Page 8, line: 375: Instead of the word "had" it is better to use the word "was characterized by".
- Page 2, lines: 43-61: In my opinion, it is worth mentioning in the Introduction the amount of the main fatty acids present in a given oil.
- Page 2, lines: 84-86: Will these fatty acid-enriched cocktails be directly consumable or do they need to be processed?
- Pages: 3 and 4, lines: 95-98: Were these lipases immobilized?
- Page: 4, lines: 147-148: Why only one catalyst ( R. miehei lipase) was chosen for further research?
- Page: 4, lines: 154-155: Why were these hydrolysis conditions used?
- Page: 6 , table 1: Why is the inscription: “ rugosa” bold?
- Page: 6 , lines: 257-260: On what basis were such hydrolysis conditions selected (activity ranges, temperature, incubation time)?
- Page: 6 , table 1: Why the results for the for almond and menhaden fish oils are given after a different incubation time than for other oils?
- Pages 8, lines: 337-339 - Why only one lipase was selected for further experiments. In my opinion, it would be good to compare the obtained results with those related to the activity of non-specific lipase?
- Page: 9, Figure 1: Are you sure the C18:1 content is so low in enzyme-free rapeseed oil and olive oil? Is there any explanation for such a low level of this fatty acid compared to other fatty acids?
- Page: 9, Figure 1: Is there any explanation for such a large increase in C18:3 in R. miehei lipase treated linseed oil? Is it not related to the distribution of fatty acids in the triacylglycerol molecules and to the specificity of the enzyme?
- Page: 9, Table 2: Why detailed results of free fatty acid concentration have only been presented for these two oils?
- Page 10, line 429: Do free fatty acids always have antioxidant properties? Does it not change during storage?
- Page: 12, line 475-476: Could you please explain where the antimicrobial activity of free acids come from?
Reviewer 3 Report
The manuscript entitled “Hydrolysis of vegetable and fish oils by fungal lipases: an effective tool to produce bioactive cocktails with antioxidant and antimicrobial potentials”
Authors: Alexandra Kotogán , Zsófia Terézia Furka , Tamás Kovács , Bettina Volford , Dóra Anna Papp , Mónika Varga , András Szekeres , Tamás Papp , Csaba Vágvölgyi , Keshab Chandra Mondal , Erika Beáta Kerekes , Miklós Takó
Overview and  genera l recommendation:
The paper investigates the hydrolysis of olive, rapeseed, linseed, almond, peanut, grape seed and menhaden oils with commercial lipases of Aspergillus niger, Rhizopus oryzae, Rhizopus niveus, Rhizomucor miehei and Candida rugosa. The main conclusion is that the lipid mixtures obtained can be reliable sources of extractable fatty acids with health-benefits. This knowledge can provide valuable information about the effects of hydrolysis of edible oils with commercial lipases to produce valuable fatty acids with antioxidant and antibacterial action. The manuscript's subject is very interesting, and the topic addressed is of great interest for the potential readers of the journal and fits within its scope. The manuscript is prepared professionally. It includes a well-written abstract and an exhaustive introduction that justifies the research undertaken. The introduction points to the deficiencies in the literature on the subject. The aim is clearly defined. The methods are standard and well described. The discussion of the results is well prepared, with comprehensive discussion. The conclusions are well-defined. The illustrative material is appropriate.
Below I give my concerns, that need revision.
The text of the manuscript should be written in passive voice. It is inappropriate to use the terms: “We concluded… We note that… In the present study, we found that… Our findings…”;
Introduction
Page 1-2 “For instance, the 43 palmitic acid (PA; C16:0), palmitoleic acid (PLA; C16:1), stearic acid (SA; C18:0), oleic acid (OA; C18:1), linoleic acid (LA; C18:2), and α-linolenic acid (ALA; C18:3) concentrations in 45 olive oil are considerable [5].” The content in this sentence is not strict. It is not true that olive oil is a source of alpha linolenic acid.
Line 43-61 Information on fatty acid composition and use of individual oils is not presented in the same frame. This should be harmonized. For all oils, it is necessary to indicate how many% of which acids are, what their health effects are, or what their use is - what they are used for. You have to organize the content.
Line 54 “Free fatty acids are presented at 2-3% in rapeseed oil [10].” Why is it only specified in rapeseed oil how many free fatty acids are there? This value (very large) applies to refined or cold pressed oil? Most often it is close to zero, especially in refined rapeseed oil.“
"However, less attention has been paid on the bioactive properties, i.e., antioxidant and antimicrobial capacities, of the produced fatty acid-enriched cocktails.”What do the authors mean by the term “cocktails”? What is the safety of eating such "cocktails"? What is their storage life? Have their effects on health been studied?
Materials and Methods
Line 98-99: „Seven oils, i.e., olive, 98 linseed, peanut, grape seed, rapeseed, almond and menhaden fish oils, were involved to 99 the assay. All oils were purchased from Sigma–Aldrich (Munich, Germany).” What were those oils? cold pressed or refined? how many free fatty acids did they have at the beginning?
Line 168: GC-MS analysis of fatty acid methyl esters – “The fatty acid composition of lipase treated and untreated oils was analyzed by GC-MS. ”A suitable method would be to determine the composition with the GC-FID method. Was the composition or content of fatty acids determined? this information is missing in the methodology. In what units are the results expressed? Usually the composition is given in%, but this is not the case here, because the content was indicated, but the percentages were not given. Please provide the relevant information in the methodology and complete the results with the acid composition in% of the individual oils.
Results and Discussion
The increase in the content of individual fatty acids after enzymatic hydrolysis of oils was determined. In my opinion, missing is: how many% of free fatty acids were? what was the fatty acid composition of the oils before and after the test? This is very important information.
Throughout the work, cocktails are mentioned only in the title and purpose. In my opinion, it is necessary to withdraw from this - especially in the title of the work, because nothing else has been written about them and this misleads the reader.
In Table 4, it would be useful to specify which of the given coefficients are significant and at which p?
Conclusions
The conclusions are an extensive summary of the entire work. In my opinion, specific brief information on what was obtained and what follows from it should be given.
Round 2
Reviewer 2 Report
Dear Authors,
Thank you very much for answering questions and making changes according to my suggestions.
At the comment 12 I was more concerned with emphasizing that the type and amount of free fatty acids after hydrolysis depends not only on the composition, but also on the distribution of fatty acids in the triacylglycerol molecules.
This remark does not have to be taken into account.
Best regards
Joanna